# Fatal Hemoptysis Secondary to Severe Pulmonary Veins Stenosis and Fibrosing Mediastinitis following Radiofrequency Ablation for Atrial Fibrillation: A Case Report and Review of the Literature

**DOI:** 10.3390/reports7010002

**Published:** 2023-12-26

**Authors:** Vladut Mirel Burduloi, Flavia Catalina Corciova, Gabriela Dumachita Sargu, Raluca Ozana Chistol, Alexandra Cristina Rusu, Cristinel Ionel Stan

**Affiliations:** 1Faculty of Medicine, “Grigore T. Popa” University of Medicine and Pharmacy, 700115 Iasi, Romania; vladut.burduloi@umfiasi.ro (V.M.B.); raluca-ozana.chistol@umfiasi.ro (R.O.C.); cristinel.stan@umfiasi.ro (C.I.S.); 2Railway Clinical Hospital, 700506 Iasi, Romania; 3“Prof. Dr. George I.M. Georgescu” Cardiovascular Diseases Institute, 700503 Iasi, Romania; flaviaantoniu@yahoo.com; 4“Elena Doamna” Obstetrics and Gynecology Clinical Hospital, 700398 Iasi, Romania; 5Faculty of Medicine, University of Medicine, Pharmacy, Science, and Technology of Targu Mures, 540142 Targu Mures, Romania

**Keywords:** pulmonary veins, stenosis, near-occlusion, cardiac CT angiography, radiofrequency ablation, atrial fibrillation

## Abstract

Fatal hemoptysis secondary to severe pulmonary veins stenosis and fibrosing mediastinitis is an exceptional late complication of radiofrequency ablation for atrial fibrillation. We report the case of a 53-year-old male with a history of atrial fibrillation treated by radiofrequency ablation and admitted in our center 6 months after the procedure because of aggravating dyspnea and fatigability. Transthoracic echocardiography showed moderate dilation of right heart cavities, severe pulmonary hypertension and a turbulent flow in superior pulmonary veins. The cardiologist suspected pulmonary vein(s) stenosis and so cardiac computed tomography (CT) angiography was performed, with findings of severe stenosis of the right superior, right inferior and left inferior pulmonary veins, near-occlusion of the left superior pulmonary vein and the vein draining the apical segment of the right lower lobe. The CT scan also revealed soft tissue attenuation of the mediastinum posterior to the left atrium suggesting fibrosing mediastinitis together with parenchymal findings consistent with pulmonary veno-oclusive disease and an area of hemorrhagic infarction. Fatal hemoptysis occurred 3 days later, before treatment was attempted. In conclusion, severe pulmonary vein stenosis and fibrosing mediastinitis are rare but devastating complications of radiofrequency ablation. Prevention and early diagnosis are the key elements as these entities are potentially life-threatening.

## 1. Introduction

Atrial fibrillation (AF) stands as the prevailing tachyarrhythmia associated with an increased risk of ischemic stroke and heart failure leading to a mortality rate of 22.3 per 100,000 population aged 35 to 84 years [1].

The pulmonary veins (PVs) play a pivotal role both in the initiation and persistence of AF. While pulmonary vein isolation (PVI) has been established as an efficacious AF treatment, it is linked to the emergence of PV stenosis, a potentially fatal complication.

Pulmonary vein stenosis defined as a ≥20% narrowing of the pre-ablation pulmonary venous diameter [2] is a condition often arising as a consequence of radiofrequency ablation (RFA) used to treat atrial fibrillation (AF). At the beginning of the 21st century, when AF ablation was in its pioneering stage, some degree (mild < 50% or moderate 50–70%) of PV stenosis occurred in up to 42% of cases [3] but severe stenosis (>70%) was infrequent (1.3%) [4]. Since then, the location of PV isolation moved from the ostium to the antral region and the technique has also evolved with optimization of the radiofrequency energy, emergence of newer methods (e.g., cryoballoon ablation) and usage of 3-dimensional electroanatomical mapping systems based on pre-procedural computed tomography (CT) imaging of the left atrium (LA) and PVs, thus decreasing the likelihood of PV stenosis. Currently, the prevalence is less than 0.3% [5]. 

Fibrosing mediastinitis, on the other hand, is an uncommon life-threatening condition not recognized as a potential complication of RFA for AF but also associated with PV stenosis. 

While most patients with PVs stenosis secondary to RFA are asymptomatic, in severe cases, it can lead to a range of symptoms like dyspnea, coughing and hemoptysis due to the gradual narrowing and occlusion of the veins and the resulting congestion in the lungs with pulmonary hypertension and venous infarction. These symptoms typically manifest gradually over a span of weeks to months following the procedure [6]. If left untreated, severe forms could prove fatal.

## 2. Detailed Case Description

We report the case of a 53-year-old male patient with a history of paroxysmal atrial fibrillation, class 1 obesity, type 2 diabetes mellitus treated with oral antihyperglycemic agents and arterial hypertension who underwent radiofrequency ablation of the cavotricuspid isthmus (CTI) and pulmonary veins under general anesthesia in another hospital two years after the initial diagnosis. 

Medical imaging prior to ablation (transthoracic echocardiography—TTE) mentioned concentric left ventricular hypertrophy with a left ventricular ejection fraction (LVEF) of 55% and no kinetic anomalies, mild aortic and tricuspid regurgitation, no right ventricular or atrial dilatation, and no anomalies of PVs. No information concerning preprocedural CT imaging of the LA and PVs was found in patient’s files. 

The radiofrequency ablation procedure was performed under general anesthesia and transesophageal echocardiographic (TEE) monitoring. The catheters (Advisor™ HD Grid Mapping Catheter, 4 mm FlexAbility™ Irrigated Ablation Catheter) were introduced via the right femoral vein and double puncture of the interatrial septum (fluoroscopic and TEE guidance). An anatomical 3D map of the left atrium was obtained using the mapping catheter and the 3D Ensite™ system with no mention of fusing the obtained map with 3D CT LA and PVs images. Circumferential radiofrequency ablation of all PVs was performed with 50 W and a duration of 10 s posteriorly and 15 s anteriorly with an irrigation flow of 17 mL/h. In a second step, the ablation of the CTI was performed with 8 applications of 40 W each. 

The discharge letter mentioned difficult positioning of the decapolar catheter with no other periprocedural accidents or incidents. The procedure was reported as successful with persistent sinus rhythm. No prior fibrosing mediastinitis was indicated in the medical documents provided by the patient. 

The patient has consistently taken anticoagulant drugs (apixaban 5 mg twice daily) as prescribed since the procedure together with oral antihyperglycemic agents, antilipidemic and antihypertensive drugs. 

The patient reported feeling well after the procedure and an insidious installation of fatigability and dyspnea after 3 months. Transthoracic echocardiography performed at 1 and 3 months after the procedure indicated a pulmonary arterial pressure of 30 mmHg. The symptoms gradually worsened, and the patent was admitted for daycare diagnostic and treatment to our clinic at approximately 6 months after the initial procedure.

Upon admission, the patient was in sinus rhythm with no atrial fibrillation episodes at Holter monitorization. Transthoracic echocardiography showed moderate dilation of right heart cavities, severe pulmonary hypertension (70 mmHg), and a turbulent flow in superior pulmonary veins (Figure 1). Pneumological evaluation revealed a mild restrictive dysfunction with a blood oxygen level of 95%. As the patient was under treatment with Eliquis^®^ (apixaban), pulmonary embolism was not suspected. Given his history of radiofrequency ablation for atrial fibrillation, the cardiologist scheduled the patient for a cardiac CT angiography to exclude a post-procedural stenosis of the pulmonary veins. 

The CT study revealed stenosis of all four pulmonary veins (severe stenosis of the right superior, right inferior and left inferior PVs, near-occlusion of the left superior PV and of the vein draining the apical segment of the right lower lobe) (Figure 2).

The patient also presented bilateral pleural effusion (40 mm on the right side and 7 mm on the left side) and a soft tissue attenuation obliterating normal mediastinal fat planes and encasing both the PVs and the esophagus posterior to the left atrium, an aspect suggesting fibrosing mediastinitis probably caused by the severe inflammation induced by radiofrequency ablation associated thermal injury (Figure 3). As for the parenchymal findings, a global mosaic attenuation was present, especially in the right middle lobe, together with a ground-glass opacity in the apical segment of the left lower lobe. The parenchymal findings were mostly consistent with pulmonary veno-oclusive disease and an area of hemorrhagic infarction.

The case was discussed by the Heart Team including cardiologists, cardiovascular surgeons and interventional cardiologists, and the decision to attempt balloon angioplasty and stenting of the PVs was made. Surgical repair was not considered an option because of the severe fibrosing mediastinitis encasing mediastinal structures around the left atrium. Also, the administration of corticosteroid treatment was considered unwarranted due to the pulmonary veins being already completely occluded.

The patient asked to be discharged until the planned procedure and developed fatal hemoptysis 3 days after the presentation during a physical effort. The ambulance report indicates that the patient was found intensely dyspneic with hemoptysis and developed irreversible cardiac arrest after being intubated in the ambulance.

## 3. Discussion

Atrial fibrillation (AF) is the most common cardiac arrhythmia with an indication for radiofrequency ablation (RFA). While generally considered safe and efficacious when performed by skilled and experienced practitioners, the procedure still carries a significant risk of serious complications, such as the development of PV stenosis, stroke, phrenic nerve palsy or atrioesophageal fistula [7]. Among these, PV stenosis is the most frequent (239 studies were published in the period 2013–2023 and were identified when interrogating the PubMed database using the keywords “pulmonary vein stenosis radiofrequency ablation”), but its real incidence is not known as the majority of patients are asymptomatic and routine cardiac CT angiography is not performed during follow-up (not mandated by the Heart Rhythm Society consensus statements) [7]. A large study performed by Teunissen et al. on 1167 patients who benefited from PV radiofrequency isolation in the period 2005–2016 reported that 36.3% of patients showed a certain degree of PV stenosis. Severe PV stenosis defined as a reduction of >70% in both superoinferior and anteroposterior diameter was noticed in only 0.7% of cases [8]. The incidence of PV stenosis decreased in the later years because of increased experience and technological progress (cryoballoon ablation, pulsed field ablation, circumferential ablation and antral isolation), Xuan et al. reporting a 2.44% overall incidence in 2020 [9]. 

According to El Baba et al. [10], the pathogenic process of PV stenosis implies a thermal injury induced by the application of energy inside the veins and inappropriate energy delivery leading to intimal proliferation, thrombosis, myocardial sleeve necrosis with collagen replacement, endovascular contraction and proliferation of elastic lamina resulting in congestion and stasis. Alaeddini et al. stated that microbubble formation is also possible in the first 2 s after starting RFA with increased thrombotic risk [11]. Stenosis of PVs can lead to infarction of the drained region and increased pulmonary capillary pressure which, in turn, has the potential to trigger lung oedema, pulmonary hypertension and subsequent right-heart failure as in our patient. Additionally, in cases of hemoptysis, PVs stenosis may cause dilation of the supplying artery and create a diversion, leading to the shunting of blood between the bronchial artery and pulmonary circulation [12]. 

In our patient, the 3D map of the LA and PVs was obtained with the mapping catheter alone without fusing the reconstructed PVs anatomy with 3D CT PVs images. This could have led to an inaccurate delineation of the LA-PVs junction and an application of RF deep inside the PVs with subsequent lesions leading to stenosis. According to Szczepanek et al., the length of the myocardial sleeve varies widely between patients and between the PVs of the same patient, with the left superior PV generally having the longest sleeve [13]. These variations may alter LA and PVs 3D map obtained with the mapping catheter if it is not adjusted after fusing with 3D CT reconstructions. We advocate for mandatory preprocedural CT evaluation of the LA and PVs to avoid imprecise delineation of anatomical features and potential complications. 

Patients with mild (30–50% narrowing) and moderate (50–70% narrowing) PV stenosis are generally asymptomatic, and only a few patients with severe stenosis of one PV develop symptoms (1 in 7 patients according to Teunissen et al.) like cough, dyspnea, chest pain, decreased exercise tolerance, hemoptysis and recurrent lung infections [8]. The occurrence of symptoms depends on the number of affected PVs, severity of PV stenosis, presence of collateral vessels and clinical setting [8]. Cardiologists should be aware of this potential complication and exclude it in all patients with a history of radiofrequency ablation presenting with the aforementioned symptoms weeks to months after the initial procedure. 

Pulmonary vein stenosis may evolve towards total occlusion, pulmonary hypertension and irreversible parenchymal damage and even prove fatal as in our patient [14]. 

The occurrence of both severe PV stenosis and fibrosing mediastinitis following RFA for AF has only been reported once by Makhija et al. in 2009 [15], with RFA not yet being a recognized potential cause of fibrosing mediastinitis [16]. Fibrosing mediastinitis is generally considered as an atypical immune overactive response within the mediastinum in reaction to specific triggering factors such as infections (histoplasmosis, tuberculosis, aspergillosis, etc.), although most cases are idiopathic. Idiopathic forms are nongranulomatous and characterized by extensive proliferation of fibrous tissue compared to the mostly focal, granulomatous forms associated with infections [16]. 

The presented case is very particular, as is the single case reported with a severe stenosis of three PVs, near occlusion of the fourth PV and a collateral branch of the right inferior PV, fibrosing mediastinitis and parenchymal findings consistent with pulmonary veno-oclusive disease and an area of hemorrhagic infarction diagnosed almost 6 months after RFA of AF. The patient was intensely symptomatic, with pulmonary hypertension and a severely impaired quality of life at presentation, the lesions proving fatal within a 3 days’ time. We plead for the inclusion of fibrosing mediastinitis among the potentially fatal complications of RFA for AF and for considering preprocedural CT evaluation of the LA and PVs as a mandatory examination before any RFA procedure. 

Further research is mandatory as fibrosing mediastinitis complicates any potential therapeutic management of PV stenosis following RFA. 

The treatment of PV stenosis is not stipulated in guidelines, and generally includes balloon angioplasty and stenting with a high restenosis rate of 24% [17]. Our patient was also scheduled for balloon angioplasty and stenting but succumbed because of fatal hemoptysis. Fender et al. [9] underline the complexity of PV stenosis treatment and recommend attempting such interventional procedures only in experienced centers. Surgical treatment is generally performed in primary PV stenosis as part of congenital heart disease [18] and not in iatrogenic-induced lesions with severe fibrosis. The most important aspect of the treatment remains prevention by careful technique selection, ablation outside PV ostia and adequate energy application.

## 4. Conclusions

Severe pulmonary vein stenosis and fibrosing mediastinitis are rare but potentially devastating complications of radiofrequency ablation imposing complex management. Prevention is the key element together with an early diagnosis as these entities could prove life-threatening.

## Figures and Tables

**Figure 1 reports-07-00002-f001:**
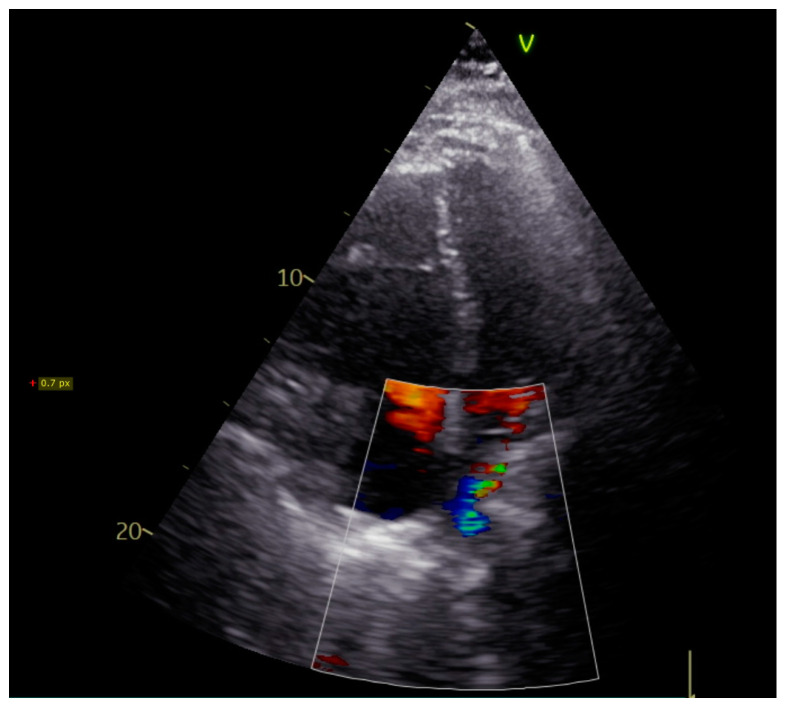
Transthoracic echocardiography showing turbulent flow in the distal part of the right superior pulmonary vein.

**Figure 2 reports-07-00002-f002:**
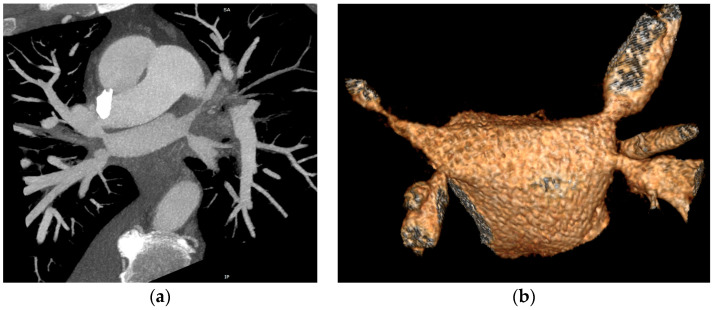
Cardiac CT angiography revealing stenosis of all four pulmonary veins: (**a**) maximum intensity projection (MIP) reconstruction; (**b**) volume rendering technique (VRT) reconstruction.

**Figure 3 reports-07-00002-f003:**
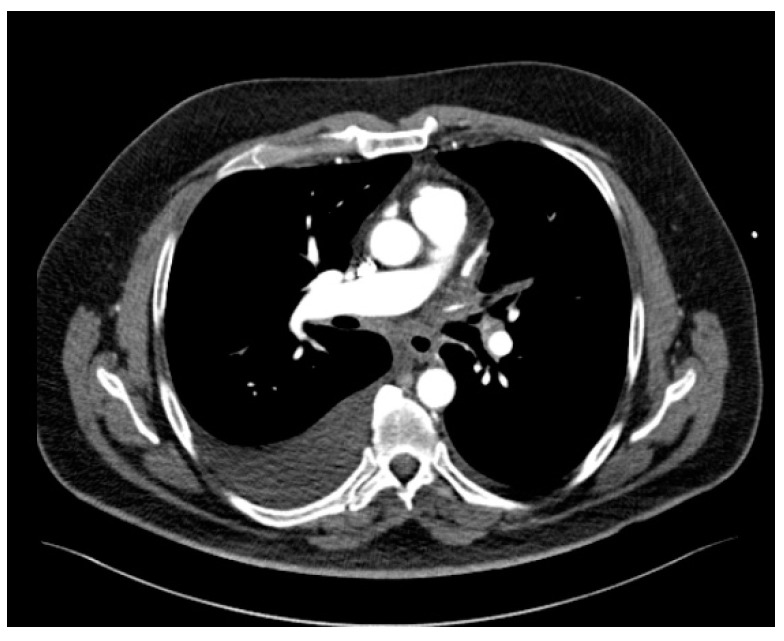
Axial CT image of the mediastinum revealing soft tissue attenuation around the esophagus suggestive for fibrosing mediastinitis.

## Data Availability

The data presented in this study are available on request from the corresponding author. The data are not publicly available due to privacy.

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
