# Peer review of "Fatal Hemoptysis Secondary to Severe Pulmonary Veins Stenosis and Fibrosing Mediastinitis following Radiofrequency Ablation for Atrial Fibrillation: A Case Report and Review of the Literature"

_reports, 2023, doi:10.3390/reports7010002_

Round 1

Reviewer 1 Report

Comments and Suggestions for Authors

In the current case report, the authors presented a patient with AF who underwent RFA and developed PV stenosis / posterior mediastinitis after the procedure. The issue is not novel. There was no data about the previous ablation procedure. Please mention the ablation settings (power, duration) and lesion approach (WACA or circling each PV?). What about pulmonary artery pressure change over time after ablation at 1, 3, and 6-month follow-up visits? Please make sure to include the details of the PV stenting procedure accordingly. There was no clinical implication or possible mechanisms for such an extensive PV stenosis in the current patient.  

Comments on the Quality of English Language

No comment

Author Response

Dear Sir/Madam,

Thank you for taking your time to analyze our manuscript. Your observations reveal thorough knowledge of the subject. We have considered your suggestions and included a detailed description of the procedure (lines 68-82). We also mentioned that the reports of 1 and 3-month TTE indicated a pulmonary arterial pressure of 30 mmHg which increased to 70 mmHg at 6 months (presentation to our hospital) (lines 91-92). Severe inflammation evolving towards fibrosis could explain the late onset of symptoms.

There was no stenting, the patient died before any treatment was attempted. The stenting was scheduled 7 days after presentation and the patient asked to be discharged until the intervention to solve some family matters. He attempted a physical effort at home and died 3 days later.

Reviewer 2 Report

Comments and Suggestions for Authors

The authors presented a case of Pulmonary vein stenosis and hemoptysis after radio frequency ablation 

The case is interesting but needs revision 

Please remove dates and replace them by time periods. Dates are identifiers

Give more details about radio ablation 

Please describe what treatment was tried for hemoptysis

Comments on the Quality of English Language

Some minor changes can be done

Author Response

Reviewer 2

Dear Sir/Madam,

Thank you for taking your time to analyze our manuscript. Your observations reveal thorough knowledge of the subject. We have considered your suggestions and replaced dates by time periods and also included technical details about the ablation procedure (lines 73-82). No treatment was tried for hemoptysis as he developed cardiac arrest after intubation in the ambulance.

Reviewer 3 Report

Comments and Suggestions for Authors

1. Lack of pre-operateive examinations such as echocardiography and chest CT scan. That's important because we need to know if the dilation of right heart and PAH had already exsisted before ablation.

2. The patient has been underwent the AF ablation, why he received long-term anticoagulation therapy. Any indications?

3. Peripheral inflammation response of the heart following catheter ablation of AF is not uncommon. However, it is mostly transient rather than chronic. Furthermore, the fibrosing mediastinitis may not be directly associated with the PV stenosis, which was more common in inappropriate ablation stratage such as the ablation target is too deep (as CTA shows). 

4. It takes some time for PV presents stenosis after ablation, as well as for subsequently develops to PAH, dilation of right heart. This part need to discuss and cite relevant references.   

Author Response

Reviewer 3

Dear Sir/Madam,

Thank you for taking your time to analyze our manuscript. Your observations reveal thorough knowledge of the subject. We have considered your suggestions and included preoperative details as right heart dilation and pulmonary artery hypertension did not exist prior to ablation (lines 68-72).

He received oral anticoagulation in case of subclinical episodes of paroxysmal AF.

We suspect that the ablation was too deep in the pulmonary veins, beyond the myocardial cuff and this was the cause of the stenosis. Fibrosing mediastinitis following RFA was previously reported by Makhija et al. 2009 and we found no other case.

We extended the discussions and added 3 more references (lines 153-163, 164-173, 200-201).

Reviewer 4 Report

Comments and Suggestions for Authors

In this case the authors report fatal hemoptysis secondary to severe pulmonary vein stenosis and fibrosing mediastinitis as a late complication of radiofrequency ablation for atrial fibrillation. TTE and cardiac CT angiography revealed findings of severe stenosis of the right superior, right inferior, and left inferior pulmonary veins and near occlusion of the left superior pulmonary vein. The CT scan also suggested fibrosing mediastinitis. Fatal hemoptysis was attempted before scheduled treatment. The authors conclude that prevention and early diagnosis of severe pulmonary vein stenosis are necessary in patients undergoing radiofrequency P.V. ablation.

The report is interesting, but could be improved:

-ETT imagines and reports data

-The inclusion criteria for the literature review are unclear. Authors should specify which databases were searched to identify eligible studies. What words have been used as keywords on a formal database search engine? If journal guidelines allow, authors may consider including a PRISMA-style diagram to clarify the inclusion/exclusion of studies.

Comments on the Quality of English Language

 Minor editing of English language required

Author Response

Dear Sir/Madam,

Thank you for taking your time to analyze our manuscript. Your observations reveal thorough knowledge of the subject. We have considered your suggestions and included preprocedural imaging reports and the RFA protocol (lines 68-82). We do not have the actual images as those examinations were performed in another hospital. The patient only provided discharge letters and consultation reports. We included the TTE images suggesting pulmonary vein stenosis at presentation in our clinic.

We cannot include a PRISMA-style diagram and a systematic review in a case presentation. We attempted that with a previous case series and the editor asked us to remove the systematic review part as it is a different type of article. We only selected large studies from those identified when interrogating PubMed database for “pulmonary vein stenosis radiofrequency ablation”.

Round 2

Reviewer 1 Report

Comments and Suggestions for Authors

The authors reasonably replied to my previous queries. I don't have any more comments. The issue has low scientific novelty. However, the concept was reviewed clearly.   

Reviewer 2 Report

Comments and Suggestions for Authors

The authors responded to the previous questions. I have no more comment,